

# RNA interference as a gene silencing tool to control *Tuta absoluta* in tomato (Solanum lycopersicum)

Roberto A. Camargo[1,2], Guilherme O. Barbosa[3], Isabella Presotto Possignolo[1,2], Lazaro E. P. Peres[2], Eric Lam[4], Joni E. Lima[1,5], Antonio Figueira[1] and Henrique Marques-Souza[3]

[1] Centro de Energia Nuclear na Agricultura, Universidade de São Paulo, Piracicaba, São Paulo, Brazil
[2] Escola Superior de Agricultura "Luiz de Queiroz" (ESALQ), Universidade de São Paulo, Piracicaba, São Paulo, Brazil
[3] Departamento de Bioquímica e Biologia Tecidual, Universidade Estadual de Campinas, Campinas, São Paulo, Brazil
[4] Department of Plant Biology & Pathology, Rutgers, The State University of New Jersey, New Brunswick, NJ, United States
[5] Departamento de Botânica, Universidade Federal de Minas Gerais, Belo Horizonte, Minas Gerais, Brazil

Corresponding authors
Antonio Figueira,
figueira@cena.usp.br
Henrique Marques-Souza,
hmsouza@g.unicamp.br

## ABSTRACT

RNA interference (RNAi), a gene-silencing mechanism that involves providing double-stranded RNA molecules that match a specific target gene sequence, is now widely used in functional genetic studies. The potential application of RNAi-mediated control of agricultural insect pests has rapidly become evident. The production of transgenic plants expressing dsRNA molecules that target essential insect genes could provide a means of specific gene silencing in larvae that feed on these plants, resulting in larval phenotypes that range from loss of appetite to death. In this report, we show that the tomato leafminer (*Tuta absoluta*), a major threat to commercial tomato production, can be targeted by RNAi. We selected two target genes (*Vacuolar ATPase-A* and *Arginine kinase*) based on the RNAi response reported for these genes in other pest species. In view of the lack of an artificial diet for *T. absoluta*, we used two approaches to deliver dsRNA into tomato leaflets. The first approach was based on the uptake of dsRNA by leaflets and the second was based on "*in planta*-induced transient gene silencing" (PITGS), a well-established method for silencing plant genes, used here for the first time to deliver *in planta*-transcribed dsRNA to target insect genes. *Tuta absoluta* larvae that fed on leaves containing dsRNA of the target genes showed an ∼60% reduction in target gene transcript accumulation, an increase in larval mortality and less leaf damage. We then generated transgenic 'Micro-Tom' tomato plants that expressed hairpin sequences for both genes and observed a reduction in foliar damage by *T. absoluta* in these plants. Our results demonstrate the feasibility of RNAi as an alternative method for controlling this critical tomato pest.

## INTRODUCTION

The mechanism of RNA interference (RNAi), in which small RNAs can rapidly cause post-transcriptional specific gene silencing, has become a powerful tool for analysing gene function in a variety of organisms (*Hannon, 2002*). The mediators of sequence-specific mRNA degradation are ~21 nucleotide-long short-interfering RNA molecules (siRNA) generated from Dicer cleavage of longer double-stranded RNA (dsRNA) (*Zamore et al., 2000*). Gene silencing by introducing dsRNA into organisms has proven to be an excellent strategy for reducing specific gene expression in several insect orders, including Diptera, Coleoptera, Hymenoptera, Orthoptera, Blattodea, Lepidoptera and Isoptera (*Katoch et al., 2013*). The potential application of RNAi-mediated control of agricultural insect pests has rapidly become evident (*Gordon & Waterhouse, 2007*), but a major challenge has been the development of an easy, reliable method for dsRNA production and delivery. The original RNAi studies used microinjection to deliver dsRNA into insects; 'soaking' has also been used as a delivery method to target cell cultures or individual larvae (*Price & Gatehouse, 2008*). The demonstration that dsRNA uptake through ingestion was sufficient to reduce target gene expression allowed the possibility of applying RNAi on a larger scale (*Koch & Kogel, 2014*). Transgenic plants engineered to express insect dsRNAs emerged as a potential technology after two independent groups proved the concept of applying RNAi to control agricultural insect pests (*Baum et al., 2007*; *Mao et al., 2007*). This approach has been developed to control lepidopteran, coleopteran and hemipteran agricultural pests (*Katoch et al., 2013*; *Li et al., 2011*; *Paim et al., 2012*), including *Helicoverpa armigera* in cotton (*Mao et al., 2011*; *Mao et al., 2015*; *Qi et al., 2015*; *Chikate et al., 2016*) and tobacco (*Zhu et al., 2012*; *Xiong et al., 2013*; *Tian et al., 2015*; *Mamta, Reddy & Rajam, 2015*), *Diabrotica virgifera virgifera* in maize (*Baum et al., 2007*; *Fishilevich et al., 2016*), *Nilaparvata lugens* in rice (*Zha et al., 2011*; *Li et al., 2011*; *Yu et al., 2014*; *Qiu et al., 2016*), *Myzus persicae* (*Mao et al., 2015*; *Tzin et al., 2015*) in *Nicotiana benthamiana* (*Khan et al., 2013*; *Pitino et al., 2011*) and *Arabidopsis thaliana* (*Coleman, Pitino & Hogenhout, 2014*; *Li et al., 2015*) and *Sitobion avenae* in wheat (*Xu et al., 2014*). However, the availability of methods that allow the screening and evaluation of candidate RNAi targets is a critical requisite for developing specific and efficient RNAi-based pest control.

The tomato leafminer, *Tuta absoluta* (Meyrick), is a small neotropical oligophagous lepidopteran that attacks many solanacean species, particularly *Solanum lycopersicum* (tomato) and other species of economic importance (*Cifuentes, Chynoweth & Bielza, 2011*; *Desneux et al., 2010*). The tomato leafminer is a multivoltine species whose young larvae can damage tomato plants during all developmental stages by forming large galleries in the leaves and burrowing into stalks, shoot apex, and green and ripe fruits. *Tuta absoluta*, which can cause yield losses of up to 100% in various regions and under diverse cultivation systems, has become the major tomato pest in South America and Africa (*Desneux et al., 2010*; *Urbaneja et al., 2013*; *Campos et al., 2014*; *Tonnang et al., 2015*). This insect invaded Europe in 2006 and spread to northern Africa in 2007 (*Urbaneja et al., 2013*), where it caused extensive economic losses to growers; multiple efforts have since been made to control this pest (*Desneux et al., 2011*). To aggravate this situation, resistance
to insecticides has been reported for *T. absoluta*, making the development of alternative means of controlling this pest even more urgent (*Urbaneja et al., 2013*; *Campos et al., 2014*; *Silva et al., 2015*).

To provide an alternative method of control based on RNAi we first investigated two approaches for delivering dsRNA to *T. absoluta* larvae, a critical requisite for efficiently screening effective target genes before developing transgenic plants. Traditionally, target genes are screened and evaluated by adding dsRNA to artificial diets offered to insect larvae (*Terenius et al., 2011*; *Zhang, Li & Miao, 2012*; *Wang et al., 2015*; *Fishilevich et al., 2016*), but in the case of *T. absoluta*, there is no readily available artificial diet (*Urbaneja et al., 2013*). We therefore investigated two delivery approaches in which *T. absoluta* larvae feed on tomato leaflets containing dsRNA. In the first approach, tomato leaflets were allowed to uptake dsRNA from an aqueous solution, in a manner similar to that described for the sap-sucking *Bemisia tabaci* (*Luan et al., 2013*). The second approach was based on the transient transcription of dsRNA by the host plant after the infiltration of *Agrobacterium* cells carrying binary plasmids that expressed hairpin versions of the target gene sequences ('agro-infiltration'). Agro-infiltration was originally developed to investigate plant-virus interactions (*Bendahmane et al., 2000*) but was later adapted to functional analyses of plant genes, e.g., for assessing gene over-expression or silencing and screening for insect resistance genes (*Grimsley et al., 1987*; *Leckie & Stewart Jr, 2010*). A method designated as "*in planta*-induced transient gene silencing" (PITGS) successfully delivered hairpin-silencing constructs in wheat to determine the pathogenicity-related gene functions of *Puccinina triticina* (leaf rust) (*Panwar, McCallum & Bakkeren, 2012*). Here, we investigated whether the same rationale could be used to silence insect genes.

The target genes selected for this study were the *Vacuolar ATPase catalytic subunit A* gene (*V-ATPase*) and *Arginine kinase* (*AK*). The $H^+$-ATPase vacuolar pump, one of the most essential enzymes present in almost all eukaryotic cells, is responsible for generating energy gradients in many membranes and organelles (*Wieczorek, 1992*; *Nelson et al., 2000*; *Wieczorek et al., 2009*). *Arginine kinase* belongs to a transferase protein family that catalyzes the transfer of a high-energy phosphate group from ATP to L-arginine to yield phosphoarginine that is used for energy storage (*Bragg et al., 2012*; *Kola et al., 2015*). These genes have been previously used as RNAi targets in exploratory studies in many different insects and crop plants (*Baum et al., 2007*; *Thakur et al., 2014*; *Kola et al., 2015*); however, this is the first report of silencing the *Arginine kinase* gene in a lepidopteran species.

Our results demonstrate the viability of using RNAi to control *T. absoluta* with delivery approaches based either on dsRNA uptake by detached tomato leaflets or through transient expression after leaf agro-infiltration, followed by infestation with pest larvae. Silencing of the insect *V-ATPase* and *AK* genes after treatment with dsRNA resulted in increased larval mortality. Further, the development of transgenic tomato lines expressing the hairpin version of these genes demonstrated that this approach can adversely affect insect development and/or viability by reducing the expression of insect target genes.

## METHODS

### Biological material

The colony of *T. absoluta* was originally established from two populations provided by the Insect Biology Laboratory of the Department of Entomology at ESALQ/USP (Piracicaba, SP, Brazil) and Bayer Crop Science (Uberlândia, MG, Brazil). This colony was reared for many generations in tomato plants or leaves under laboratory conditions (25 ± 2 °C; 60 ± 10% relative humidity; 14 h photoperiod). 'Santa-Clara' or 'Cherry' tomato cultivars were used for *T. absoluta* maintenance and in dsRNA uptake experiments. The 'Micro-Tom' cultivar was used for genetic transformation.

### Target gene selection

Genes encoding a vacuolar *V-ATPase* and *AK* were chosen based on previous successful reports of RNAi used for insect control (*Baum et al., 2007*; *Zhao et al., 2008*; *Kola et al., 2015*). Since no sequences were available for *T. absoluta* genes, degenerated primers (Table S1) were developed based on conserved amino acid sequence regions from aligned homologs of *Aedes aegypti* (GenBank accession *V-ATPase*: XP_001659520.1), *Bombyx mori* (*V-ATPase*: NP_001091829.1; *AK*: NP_001037402.1), *Drosophila melanogaster* (*V-ATPase*: NP_652004.2; *AK*: AAA68172.1), *Helicoverpa armigera* (*AK*: ADD22718.1), *Heliothis virescens* (*AK*: ADE27964.1), *Homalodisca vitripennis* (*AK*: AAT01074.1), *Manduca sexta* (*V-ATPase*: P31400.1), *Spodoptera litura* (*AK*: ADW94627.1) and *Tribolium castaneum* (*V-ATPase* XP_976188.1; *AK*: EFA11419.1). Based on these orthologous genes, the complete *V-ATPase A* coding sequence was estimated to be around 1850 bp, while the *AK* coding sequence ranged from 1,062 to 1,476 bp.

### *Tuta absoluta* RNA extraction and cDNA synthesis

Total RNA was extracted from 100 mg of *T. absoluta* larvae at the four instar stages using TRIzol (Invitrogen; Carlsbad, CA, USA). RNA was quantified spectrophotometrically and analyzed by electrophoresis in 1% denaturing agarose gels in MOPS buffer. Around 1 µg of total RNA was treated with DNase I and 20 U of Ribolock (Fermentas, Burlington, Canada) at 37 °C for 30 min, with the reaction stopped by adding EDTA (50 mM) and heating to 65 °C for 10 min. One microgram of DNase-treated RNA samples was reversed transcribed in a total reaction volume of 20 µL containing 500 µM of each dNTP, 2.5 µM oligo dT, 5 mM DTT and 200 U SuperScript III (Invitrogen) in appropriate buffer at 50 °C for 60 min, followed by enzyme inactivation at 70 °C for 15 min.

### Target gene amplification and cloning

Target genes were amplified from cDNA using a nested PCR-based method with degenerate primer pairs (Table S1) in a 20 µL reaction volume containing ∼100 ng of cDNA, 3 mM MgCl₂, 100 µM of each dNTP, 1 µM of each primer and 2 U of High Fidelity *Taq* DNA polymerase (Invitrogen) in the appropriate buffer. Amplifications were done in a Veriti thermocycler (Applied Biosystems, Foster City, CA, USA) programmed to cycle at 94 °C for 5 min, followed by 35 cycles of 94 °C for 40 s, 45 °C for 60 s, 72 °C for 60 s and a final cycle at 72 °C for 10 min. The second reaction was run under the same conditions

as the first reaction using 1 μL from the latter. Amplification products were analyzed by gel electrophoresis and target fragments were excised, purified using a PureLink Quick gel extraction kit (Invitrogen) and cloned into pGEM-T Easy vector (Promega, Madison, WI, USA) using standard procedures. Identity was confirmed by sequencing three clones from each target gene in an ABI PRISM 3130 (Applied Biosystems).

## Cloning the target gene fragments as hairpins in the RNAi silencing vector

To clone the target gene fragments in the binary silencing vector pK7GWIWG2(I) (*Karimi, Inze & Depicker, 2002*) primers were synthesized to amplify fragments flanked by the recombination sequences *attL1* and *attL2* (Table S2). The amplification reactions contained 10 ng of pGEM-T-cloned DNA, 200 μM of each dNTP, 0.4 μM of each primer and 2 U of Phusion DNA polymerase (New England Biolabs, Ipswich, MA, USA) in a final reaction volume of 50 μL. Temperature cycling for amplification was programmed to start at 98 °C for 2 min, followed by 35 cycles at 98 °C for 10 s, 60 °C for 15 s and 72 °C for 15 s, with a final cycle at 72 °C for 5 min. The amplified products were purified from agarose gels using QIAEX II gel extraction kits (Qiagen, Hilden, Germany), quantified, recombined into pK7GWIWG2(I) using LR clonase (Invitrogen) according to the manufacturer's instructions and transformed into TOP10 *Escherichia coli* cells. The presence of the insert was confirmed by amplification and the direction of insertion was verified by digestion and sequencing. Successful constructs were then transformed by heat shock into *Agrobacterium tumefaciens* GV3101/MP90 cells and confirmed by PCR.

## dsRNA synthesis

pGEM-T Easy clones containing *T. absoluta V-ATPase* and *AK* gene fragments were used as a template for transcription *in vitro* to produce dsRNA using T7 RNA polymerase (MegaScript T7; Life Technologies, Carlsbad, CA, USA). Target sequences cloned into pGEM-T Easy were amplified with a T7 primer and a SP6 primer fused to a T7 sequence (TAATACGACTCACTATAGGGATTTAGGTGACACTATAG) to operate as a T7 promoter for bidirectional *in vitro* transcription. The amplification reactions contained 10 ng of plasmid DNA, 1.5 mM MgCl$_2$, 200 μM of each dNTP, 0.5 μM of each primer and 1.0 U of *Taq* polymerase in a final reaction volume of 20 μL. Temperature cycling for amplification started at 95 °C for 2 min, followed by 35 cycles of 15 s at 95 °C, 20 s at 60 °C, 30 s at 72 °C, with a final extension at 72 °C for 5 min. Amplified fragments were run on and purified from 1% agarose gels as described above. Purified products were quantified by fluorimetry and used for *in vitro* transcription reactions containing 100 ng of target DNA, 7.5 mM of each ribonucleotide and 200 U of MegaScript T7 in appropriate buffer in a final volume of 20 μL. The reactions were run at 37 °C for 16 h, followed by the addition of 2 U of DNase for 15 min at 37 °C. dsRNA was purified by precipitation with 7.5 M LiCl (30 μL) at −20 °C for 1 h followed by centrifugation (12,000 g, 15 min, 4 °C). The RNA pellet was washed with 70% ethanol and resuspended in DEPC-treated water. The green fluorescent protein (GFP) gene was used as a negative control. The vector pCAMBIA1302 was used as a template to amplify a negative control *GFP* gene fragment (276 bp) with the specific primers

GFP-F (5′- TAATACGACTCACTATAGGGCAGTGGAGAGGGTGAA) and GFP-R (5′- TAATACGACTCACTATAGGGTTGACGAGGGTGTCTC), both containing additional T7 sequences (underlined). Similar transcription *in vitro* was done with this template.

### Labeling dsRNA to follow uptake by tomato leaflets and ingestion by *T. absoluta*

*In vitro* transcription of dsRNA was done as described above, except that 2 μL of Cy3-labelled riboCTP was added (*Zhang et al., 2013*). Fluorescently-labelled dsRNA was purified using a MEGAclear kit (Ambion, Carlsbad, CA, USA) and provided in solution to tomato leaflets at 500 μg mL$^{-1}$. First instar larvae ($n = 15$) were placed on the treated leaflets and collected 6 h or 24 h later for observation by confocal fluorescent microscopy with an AxioVision Zeiss LSM780-NLO microscope (Carl Zeiss AG).

### dsRNA delivery to *T. absoluta* larvae via feeding

dsRNA was delivered into tomato leaves by two methods and the effects were evaluated based on gene expression analysis in fed larvae, larval mortality or tomato tissue damage. In the first method, detached leaflets from 'Santa-Clara' tomatoes had their petioles immersed in 200 μL of water containing either 5 μg of dsRNA from each target gene or a *GFP* control, in triplicate. Uptake of the dsRNA solution by the tomato leaflets took 3–4 h. Immediately after uptake, first instar larvae ($n = 50$–$100$) were gently placed onto leaflets for feeding and individuals were sampled 24 h, 48 h and 72 h after initiation of feeding. Negative controls with dsRNA from the *GFP* gene sequence were run in parallel. The effects of RNAi on the larvae were evaluated by quantitative amplification of reversed transcripts (RT-qPCR) of each target gene compared to the control.

The second delivery method, termed here PITIGS (Plant-induced Transient Insect Gene Silencing), was based on the ''*in planta*-induced transient gene silencing'' (PITGS) method, an established method for silencing plant genes (*Panwar, McCallum & Bakkeren, 2012*). As a proof of concept, 'Santa Clara' tomato leaflets were infiltrated with *Agrobacterium* cells containing hairpin target gene fragments cloned into the pK7GWIWG2(I) vector or a similar hairpin expression construct for the *GFP* gene as a transient assay. Initially, to validate the potential of gene silencing in a transient assay, leaves were either infiltrated only with *Agrobacterium* GV3101/pMP90 cells containing an expression construct for enhanced GFP (*eGFP*) to visualize the GFP transient expression or in combination with another *Agrobacterium* line containing a GFP silencing construct (*GFPi*). The *Agrobacterium* suspensions were infiltrated into the abaxial side of the tomato leaves using a microsyringe and the treated area was marked with a permanent marker. Leaf tissues that had been agro-infiltrated with the *eGFP* or *eGFP* plus *GFPi* constructs were examined two days after treatment using a confocal fluorescent microscope (AxioVision Zeiss LSM780-NLO; Carl Zeiss AG, Oberkochen, Germany) to monitor the degree of gene silencing based on *eGFP* expression.

In subsequent feeding assays, tomato leaflets were infiltrated with *Agrobacterium* cells carrying the *V-ATPase* hairpin constructs or with *Agrobacterium* cells carrying the *GFPi*

construct as a negative control, in triplicate. The *Agrobacterium* cells were grown on LB medium containing gentamycin ($25 \, \mu g \, mL^{-1}$) and spectinomycin ($100 \, \mu g \, mL^{-1}$) for 12 h, centrifuged at 3,000 g for 5 min and resuspended in water to an $OD_{600nm} = 0.5$. After 24 h, first instar *T. absoluta* larvae were placed on the treated leaf areas and 24 h, 48 h and 72 h later treated larvae and their respective controls were sampled for analysis of *V-ATPase* and *AK* expression by RT-qPCR.

Feeding assays were also done to estimate larval mortality. Detached 'Santa Clara' leaflets had their petioles immersed in an aqueous solution containing 1 µg of dsRNA of each target gene (*V-ATPase* or *AK,* plus *GFP* control), a procedure that was repeated daily for 10 days, with a total of 10 µg being provided to each leaflet. A total of 10 first instar larvae were placed to feed on these treated leaflets (in triplicate) and larvae mortality was estimated after 5, 7, 10 and 24 days of treatment. Under the conditions used here, the feeding cycle of *T. absoluta* lasted 10–12 days from larval emergence to pupae, and a total of *ca.* 20–24 days for adults to emerge.

An additional assay was done using 'Santa Clara' tomato leaflets with the petiole immersed in increasing amounts of dsRNA (total: 500, 1,000 or 5,000 ng) of the target genes (*V-ATPase* or *AK*) or *GFP* negative control (in duplicate). Five larvae were placed on each leaflet and leaflets were photographed for 11 days to visually assess the extent of damage.

## Quantitative amplification of reversed transcripts (RT-qPCR)

For gene expression analysis of treated *T. absoluta* larvae, cDNA was synthesized from RNA samples collected from the surviving larvae at specific time points for each experiment using a High-Capacity cDNA reverse transcription kit (Applied Biosystems) following the manufacturer's instructions, using 1 µg of DNase-treated total RNA, random primers and 50 U of MultiScribe reverse transcriptase in 20 µL. RT-qPCR reactions contained ∼40 ng of larval cDNA, 5 µL of Fast SYBR Green Master (Invitrogen) and 0.2 µM of each gene-specific primer (Table S4) in a total volume of 10 µL. Amplifications were done starting at 50 C for 10 min and 95 C for 2 min, followed by 40 three-step cycles of 95 C for 15 s, 60–61 °C for 25 s and 72 C for 30 s in a Qiagen RotorGene-6000 (Qiagen). After amplification, melting curves were determined between 72 °C and 95 °C. Reactions were done with biological replicates, technical triplicates and non-template controls. Primer efficiency was determined using a pool of cDNA in serial dilutions ($10, 10^{-1}, 10^{-2}$ and $10^{-3}$). $C_Q$ values were used to determine differences in expression based on *Livak & Schmittgen (2001)*. Reference genes were *RpL 5* (large subunit 5 ribosomal protein), *Rpl23A* (large subunit 23A ribosomal protein) and *rRNA* (Table S4). Negative controls were larvae fed on *GFP* dsRNA.

## 'Micro-Tom' genetic transformation with the RNAi silencing constructs

The *ATPase* and *AK* silencing constructs were used to transform 'Micro-Tom' based on published protocols (*Pino et al., 2010*). Cotyledon explants were obtained from 8-day old seedlings cultivated on MS medium supplemented with sucrose ($30 \, g \, L^{-1}$), B5 vitamins and

0.4 µM naphthalene acetic acid (NAA) (*Pino et al., 2010*). A single colony of *Agrobacterium* GV3101/MP90 (pK7GWIWG2(I)::*ATPase-esaPTA* or pK7GWIWG2(I)::*AK-KA*) grown for two days on 3 mL of LB medium with spectinomycin (100 mg L$^{-1}$), rifampicin (50 mg L$^{-1}$) and gentamycin (25 mg L$^{-1}$) was inoculated into 50 mL of LB medium with the same antibiotics and incubated overnight at 120 rpm and 28 °C. The suspensions were then centrifuged (1,000 g, 15 min, 20 °C) and the pellet was re-suspended in liquid MS medium containing sucrose (30 g L$^{-1}$), with the OD$_{600nm}$ adjusted to 0.2–0.3. Acetosyringone (100 µM) was added to the suspensions 10 min before co-cultivation, which was done on the same semi-solid MS medium for two days in the dark at 25 °C. Explants were then transferred to fresh MS medium supplemented with B5 vitamins, sucrose (30 g L$^{-1}$), 5 µM benzylamino purine (BAP), kanamycin (100 mg L$^{-1}$) and timetin (300 mg L$^{-1}$) and maintained on a 16 h photoperiod at 25 °C for three weeks. Subsequently, adventitious shoots >5mm long were transferred to identical medium until roots developed and the plantlets were hardened (∼two weeks), after which they were moved to a greenhouse.

## Genetic analysis of transgenic plants

Total DNA and RNA were extracted from putative transgenic plants using Trizol, by the Acid Guanidinium Thiocyanate-phenol-chloroform extraction method (*Chomczynski & Sacchi, 1987*). Confirmation of transgenesis was done by PCR using a 35S promoter sense primer (GCACAATCCCACTATCCTTC) together with a target gene (*ATPase* and *AK*)-specific reverse primer (Table S4). Reactions (final volume of 25 µL) contained 100 ng of genomic DNA, 1.5 mM MgCl$_2$, 0.2 µM of each dNTP, 0.1 µM of each primer and 1.5 U of *Taq* DNA polymerase in appropriate buffer (Fermentas). Amplification started at 95 °C for 2 min, followed by 35 cycles of 15 s at 95 °C, 25 s at 60 °C and 40 s at 72 °C, with a final extension at 72 °C for 5 min. The products were analyzed in 1% agarose gels. To confirm transcript expression in transgenic plants, RT-PCR reactions were done using gene specific primers for *ATPase* or *AK* (Table S4) together with primers for a tomato ubiquitin gene as an endogenous reference gene. Reverse transcription was done with 1 µg of DNase I-treated total RNA, 0.75 µM of gene-specific primers (*V-ATPase* or *AK*), 0.75 µM of ubiquitin primers, 0.5 mM of each dNTP and 200 U of Revertaid (Fermentas) in a final volume of 20 µL. For RT-PCR, 1 µL of cDNA, 1.5 mM MgCl$_2$, 0.2 mM of each dNTP, 0.1 µM of each primer and 1.5 U of *Taq* DNA polymerase (Fermentas) were mixed in a final volume of 25 µL and amplified at 95 °C for 2 min, followed by 35 cycles at 95 °C for 15 s, 60 °C for 30 s and 72 °C for 20 s. Amplification products were examined by gel electrophoresis.

To specifically detect siRNA expressed in transgenic plants, a stem-loop pulsed RT-PCR was done using the protocol described by *Varkonyi-Gasic et al. (2007)*. Initially, the introduced gene fragment sequences were analyzed with a software for virtual prediction of siRNA (*Rice, Longden & Bleasby, 2000*). Stem-loop primers specific for two virtually predicted siRNA for each gene, gene-specific primers and a universal primer (*Varkonyi-Gasic et al., 2007*) were obtained (Table S5). As an endogenous control, primers for microRNA156 (*MIR156*) were also developed based on sequences available at miRBASe (*Kozomara & Griffiths-Jones, 2011*). Reverse transcription reactions used 0.05 µM of stem-loop primers and 1 µg of DNase-treated total RNA that were heated to 70 °C for
10 min and chilled to 4 °C for 10 min. Subsequently, 3 mM MgCl$_2$, 0.25 mM of each dNTP and 200 U of Revertaid (Fermentas) in appropriate buffer were added to a final volume of 20 µL. The reaction was run at 16 °C for 30 min, followed by 60 cycles at 30 °C for 30 s, at 42 °C for 30 s and 50 °C for 1 s, with a final step at 85 °C for 5 min to inactivate the enzyme. Amplification was then done in reactions consisting of 1 µL of cDNA, 1.5 mM MgCl$_2$, 0.2 mM of each dNTP, 0.2 µM of each primer and 1.5 U of *Taq* DNA polymerase in appropriate buffer (Fermentas). Temperature cycling started at 94 °C for 2 min, followed by 35 cycles at 94 °C for 15 s and 60 °C for 1 min. The products were examined after separation on 3% agarose gels.

### Feeding assay using transgenic plants

Leaves from 'Micro-Tom' plants (T$_0$) transgenic lines for *V-ATPase* and *AK* and their respective controls had their petioles immersed in water. Ten recently hatched larvae were then allowed to feed on the leaves, which were photographed for seven days to monitor leaf tissue damage. Pupae were collected at the end of the larval cycle for counting and weighing.

## RESULTS

### Target gene isolation from *T. absoluta* and dsRNA transcription *in vitro*

Since little genomic information is available for *T. absoluta,* we conducted target gene fragment cloning using degenerated primers for both target genes (estimated coding sequence of ~1,850 bp for *V-ATPase* and ~1,065 bp for *AK*) by using nested PCR (Table S1). The final amplification products were run on agarose gels and fragments for both genes were purified and cloned. Three positive clones were sequenced in both directions for each target gene. The consensus sequence assembled from the three clones contained 285 bp for *V-ATPase* (GenBank: KM591219) and 262 bp for *AK* (GenBank: KM591220). The sequences were conceptually translated and aligned to homologs from other species (Fig. S1); both sequences displayed conserved domains (GenBank: cd01134 for *V-ATPase* and GenBank: cd07932 for *AK*).

In all RNAi assays, the same regions of the target and green fluorescent protein (GFP) control genes were used, either by *in vitro* transcription of the dsRNA or by transient/stable transgenesis.

### dsRNA delivery methods for *T. absoluta* in tomato
#### Leaf uptake of dsRNA and larval ingestion

To overcome the lack of a suitable artificial diet for *T. absoluta*, we used alternative methods to deliver dsRNA to the insect larvae in order to facilitate the rapid screening of candidate target genes. The first approach involved supplying dsRNA transcribed *in vitro* to tomato leaflets by petiole uptake. Detached tomato leaves absorbed a solution of dsRNA transcribed *in vitro* by the petioles and first instar larvae subsequently fed on these treated leaves. To determine whether the dsRNA could be successfully absorbed and transported to the leaf laminae and then be ingested by the insect to reach its digestive tract, Cy3-labled (red

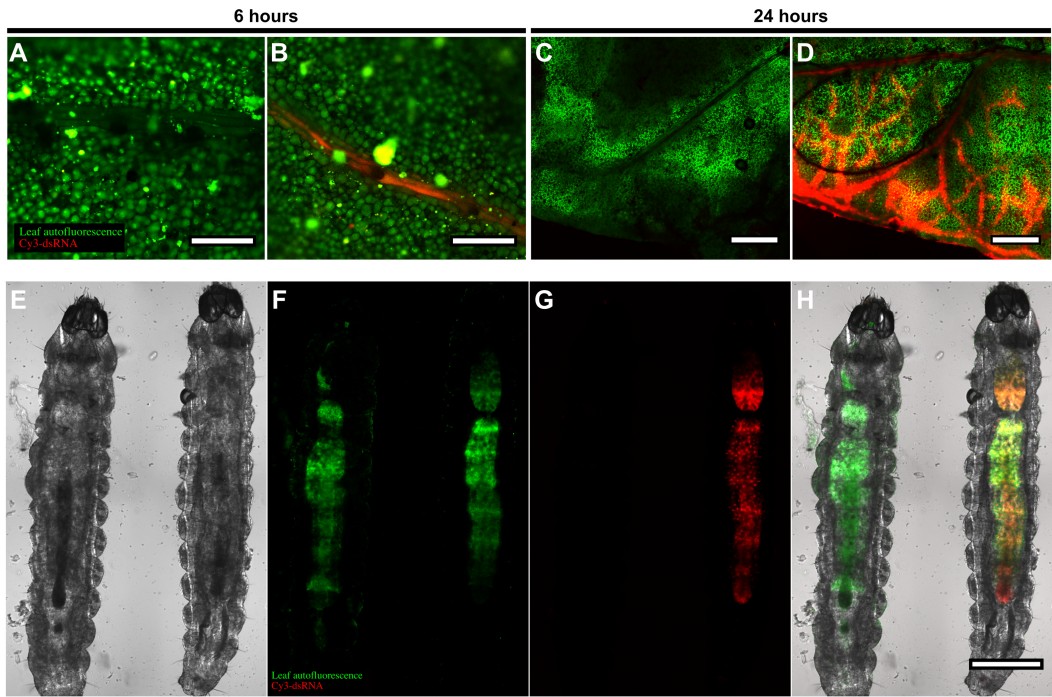

**Figure 1** **Trajectory of Cy3-labeled dsRNA molecules through the tomato leaflets up to the larvae intestinal track.** (A–D) Confocal images of tomato leaflets exposed (B, D) or not (control) (A, C) to dsRNA from V-ATPase gene fragment labeled with Cy3 (seen as red dye) by *in vitro* transcription, taken at 6 h (bar = 250 μm) (A, B) or 24 h (bar = 1 mm) (C, D) after treatments were imposed. Chlorophyll is shown as green fluorescence. (E–H) Confocal images of 1st instar Tuta absoluta larva fed on tomato leaflets exposed (larva on the right hand side) or not (control—larva on the left hand side) to dsRNA from V-ATPase gene fragment labeled with Cy3 by in vitro transcription, taken 24 h after treatments were imposed (bar = 250 μm). Larvae images were taken with bright light (E), 488 (F), 555 (G) channels and images (E), (F) and (G) were merged in H.

fluorescence) dsRNA fragments of *V-ATPase* transcribed *in vitro* were provided in solution to detached tomato leaflets.

The treated leaves and the feeding larvae were imaged by confocal microscopy 6 h or 24 h after treatment (Fig. 1, Fig. S2). Labelled dsRNA species were already strongly detected in the leaflet petiole and blade (mid-rib and lateral veins) of the leaflets 6 h after treatment (Fig. 1Ab). After 24 h, Cy3-labeled RNA molecules were detected throughout the leaf blade (Fig. 1Ad, Fig. S2). With time, Cy3-labeled RNA molecules accumulated at the leaf margin until saturation was reached in certain areas (Fig. 1Ad).

We then imaged larvae fed on treated or untreated leaflets using the 488 channel (green fluorescence) to detect chlorophyll auto-fluorescence, indicative of plant tissue ingestion by the larvae, and the 555 channel (red fluorescence) to detect Cy3 fluorescence (Fig. 1B). In both treatments, green fluorescence was detected throughout the larval digestive tract (Fig. 1Bb), indicating that the larvae fed normally under both circumstances. However, larvae fed on dsRNA-treated leaflets showed a strong Cy3 signal in the digestive tract, indicating the presence of leaflet-absorbed Cy3-labeled RNA molecules in the gastric caeca of the midgut (Fig. 1Bc).

### Plant Induced Transient Insect Gene Silencing (PITIGS)

We took advantage of the transient expression by *Agrobacterium* infiltration to develop a system to mimic stable transgenesis in which plant cells expressing hairpin versions of the target gene to transcribe the dsRNA targeting insect specific genes. This approach is based on PITGS (Plant Induced Transient Gene Silencing), an established method for silencing plant genes (*Panwar, McCallum & Bakkeren, 2012*). We named this approach PITIGS (Plant Induced Transient Insect Gene Silencing). As a proof of concept, we infiltrated 'Santa Clara' tomato leaves with two *Agrobacterium* strains: one containing an expression cassette for enhanced GFP expression (*eGFP* strain) and another in which a 400 bp fragment of *eGFP* was cloned into a binary expression vector as inverted repeats in order to transcribe a hairpin version (dsRNA) of the *eGFP* gene (*GFPi* strain).

Confocal fluorescence microscopy showed that agro-infiltrated leaves with the *eGFP* line displayed GFP (green) fluorescence as sparse cells on the leaf blade (Fig. S3A). When both *Agrobacterium* strains (*eGFP* and *GFPi*) were co-infiltrated, there was a drastic reduction in the number and intensity of cells with GFP fluorescence (Fig. S3B); this fluorescence was similar to that of leaf regions without agro-infiltration (Fig. S3C).

To employ PITIGS for the target genes, fragments of *V-ATPase* and *AK* were cloned into a binary vector as a hairpin-expressing cassette and agro-infiltrated into 'Santa Clara' tomato leaves. The cloned fragments were amplified with primers flanked by *attL1* and *attL2* sequences (Table S2) to enable direct recombination with the binary vector pK7GWIWG2(I) (*Karimi, Inze & Depicker, 2002*).

### Effect of RNAi on target gene expression

For both RNAi delivery methods, larvae were allowed to feed exclusively on RNAi-treated leaflets (dsRNA uptake or agro-infiltration) and collected 24 h, 48 h and 72 h later. The relative expression of *V-ATPase* and *AK* was quantified by RT-qPCR.

Larvae fed on leaflets treated by the dsRNA uptake delivery method showed a significant decrease in transcript accumulation for both genes 48 h and 72 h after treatment ($\sim$40% reduction at 72 h after treatment) (Fig. 2A). Larvae fed on agro-infiltration leaflets showed a decrease in transcript accumulation at all time points, with the highest decrease occurring 72 h after treatment, ($\sim$35% reduction for *V-ATPase* and 40% reduction for *AK*) (Fig. 2B).

Considering that both dsRNA delivery approaches resulted in similar gene silencing effects, subsequent experiments were done using only the leaf dsRNA uptake delivery method.

### Effect of RNAi on larval mortality

To determine the effect of RNAi on larval mortality, *T. absoluta* larvae were allowed to feed on single leaflets ($n = 3$) that had absorbed 10 µg of dsRNA from *V-ATPase*, *AK* or *GFP* prior to the start of the feeding essay. Larvae were then sampled after five, seven and ten days of treatment and an additional pupal sample was collected after 24 days. Larval mortality was significantly higher in larvae fed on leaflets that absorbed dsRNA of either target gene when compared to the *GFP* control at all time points, with an additional increase over time (Fig. 3). By day 24, mortality had reached an average of 50% for *V-ATPase* and 43%

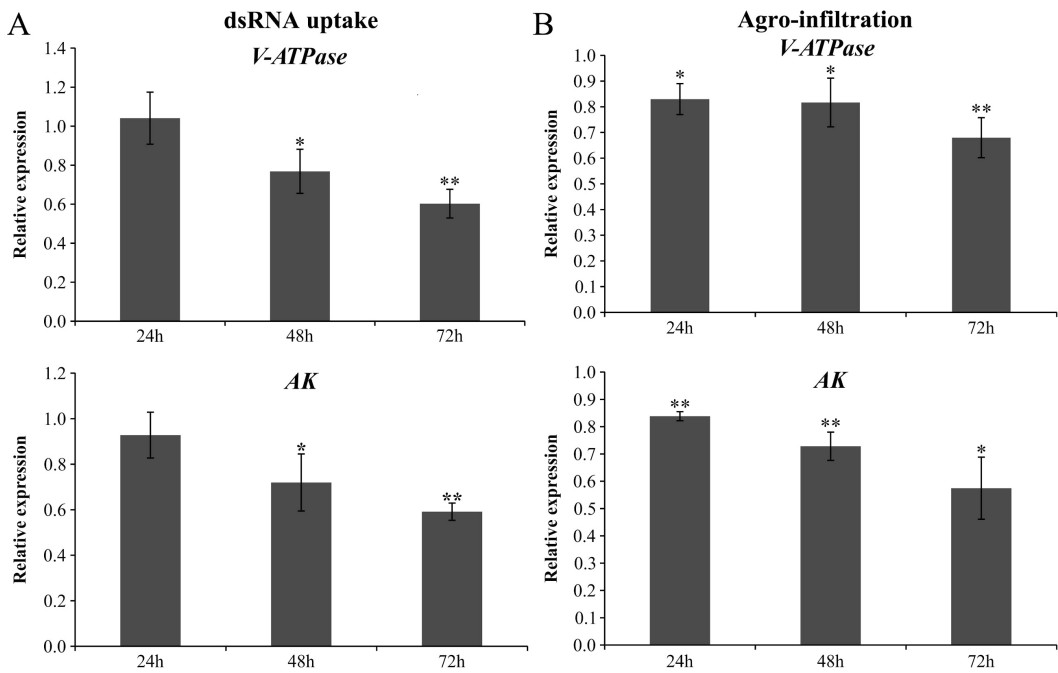

**Figure 2   Effect of the dsRNA uptake and PITIGS delivery methods on the relative expression of target genes in *T. absoluta* larvae.** Relative expression of the target genes *V-ATPase* and *AK* in *T. absoluta* larvae fed on tomato leaflets that absorbed a dsRNA solution (25 μg mL$^{-1}$) (A) and tomato leaflets that were in-filtrated with *Agrobacterium* cells containing constructs that transcribed inverted repeats of the target gene fragments *V-ATPase* and *AK* (B). The larvae in A and B were sampled 24 h, 48 h and 72 h after the initiation of feeding. Gene expression was normalized to positive controls that were exposed to *GFP* dsRNA ($n = 3$). The *Rpl 5* gene was used as a reference gene ($n = 3$). The columns represent the mean ± SE. *$P < 0.05$ and **$P < 0.01$ (Student's *t*-test).

for *AK* compared to 17% for the *GFP* control (Fig. 3). Independent experiments using different total amounts of dsRNA in the leaflets yielded similar results (not shown), with no pleiotropic effect observed when using higer concentrations of dsRNA for the target genes. Evaluation of larvae after 11 days of treatment (Figs. S4A–S4C) and at the pupal stage (Figs. S4D–S4F) revealed that larvae fed on RNAi-treated leaflets displayed developmental delay, reduced body size, external morphologies of the 3$^{rd}$ instar stage (when 4$^{th}$ instar was expected) (Figs. S4A–S4C), failure to pupate (Figs. S4D–S4F), and failure to emerge as adults (data not shown).

### Effect of RNAi on tomato leaf damage

We next assessed whether the gene silencing and larval mortality observed for both RNAi target genes resulted in less herbivory by *T. absoluta* on tomato leaves. After 11 days of exposure to *T. absoluta*, leaflet blades treated with increasing amounts of dsRNA (total: 500, 1,000 or 5,000 ng) of the target genes (*V-ATPase* or *AK*) (Figs. S4C–S4F) were visibly less damaged by larval herbivory when compared to leaflets treated with GFP dsRNA (Figs. 4A, 4B); the observed protective effect appeared to be dose-dependent (Figs. S4C–S4F). A protective effect was seen even at lower doses of dsRNA treatment, particularly for *V-ATPase* (Fig. 4D).

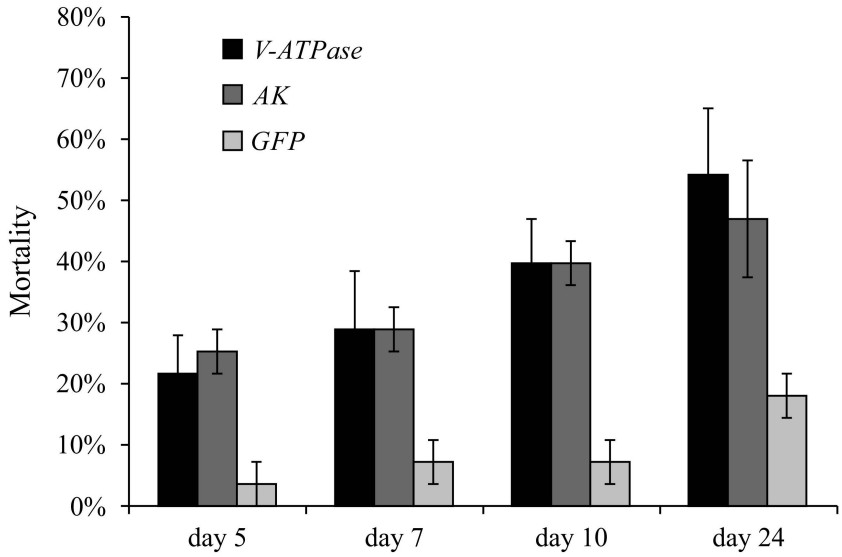

**Figure 3** **RNAi effects on larval mortality.** Mortality of *Tuta absoluta* larvae ($n = 10$) after feeding on tomato leaf treated with dsRNA from *V-ATPase*, *AK* or *GFP* control for 24 days. Tomato leaflets were provided with one μg of each dsRNA (*V-ATPase* or *AK*, plus *GFP* control) a day per leaflet for 10 days in a total 10 μg.

### RNAi transgenic tomato plants

To examine the effects of constitutive RNAi *in planta*, pK7GWIWG2(I) plasmids containing *V-ATPase* or *AK* employed in the *Agrobacterium* transient hairpin expression were used to stably transform 'Micro-Tom' tomato plants. 'Micro-Tom' plants were successfully transformed in three independent experiments (Table S3). The efficiency of transformation ranged from 3.5% to 18.9% and varied among gene constructs. To characterize the events, 13 plants derived from 11 events of the *V-ATPase* construct transformation were analyzed by PCR amplification and eight events were positive for the transformation. From *AK* constructs, nine plants from six events were characterized and all were positive. The confirmed transgenic plants (eight *V-ATPase* and nine *AK*) were analyzed for the presence of the full transcript by RT-PCR; all *V-ATPase* plants plus six *AK* plants (#2.1, #2.2, #2.3, #3, #5 and #6) showed the presence of the respective transcript (Fig. S5A). The plants were also analyzed for the presence of a potential siRNA derived from the gene fragments inserted by stem-loop pulsed RT-PCR using specific primers (Table S4), and all plants from both target genes presented the expected siRNA, except for *AK* event #4 (Fig. S5B). Absence of the transcript in *AK* #4 was also demonstrated by RT-PCR. Conversely, events *AK* #1.1 and *AK* #1.2 did not show the presence of the transcript by RT-PCR, but the stem-loop pulsed RT-PCR assay revealed the presence of the expected siRNA.

### Larval survival and leaf damage in RNAi transgenic tomatoes

To test whether RNAi transgenic plants would promote larval mortality and protect against herbivory, *T. absoluta* larvae were allowed to feed on $T_0$ transgenic 'Micro-Tom' leaves until emergence as adults. Larvae fed on these leaves were collected and larval mortality was evaluated by counting survivors and by visually detecting deleterious larval phenotypes

**Day 11**

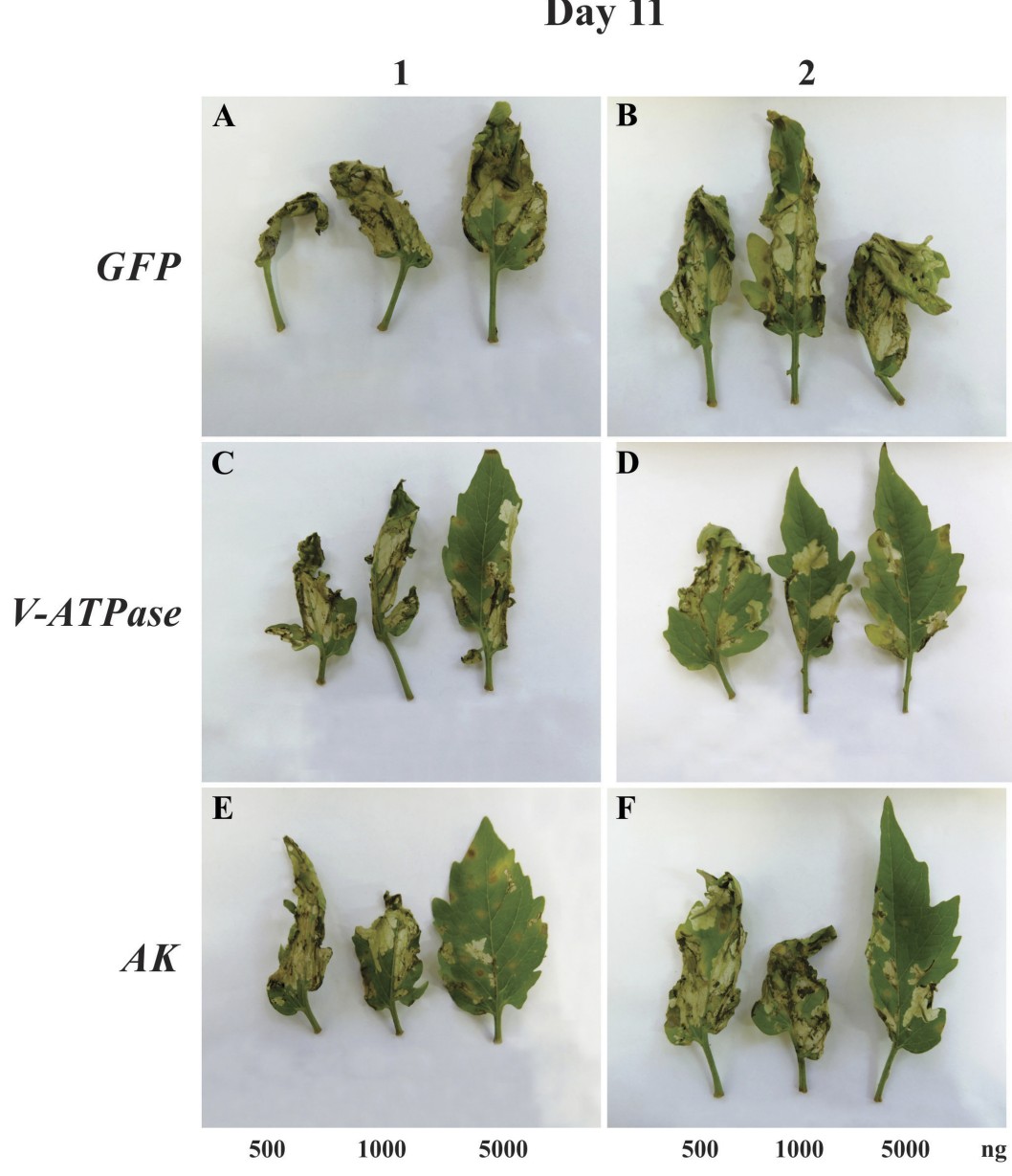

**Figure 4   RNAi effects on leaf damage caused by larval herbivory.** Tomato leaflets that absorbed increasing amounts of dsRNA (500, 1,000 or 5,000 ng) of GFP (A, B), *V-ATPase* (C, D) and *AK* (E, F), submitted to *T. absoluta* larvae herbivory for 11 days.

based on larval size and weight. When fed on non-transgenic control plants, 100% of the larvae developed normally and reached the pupal stage. Conversely, feeding on RNAi transgenic plants resulted in a significant increase in larval mortality that ranged from 30% in events ATPase1.1 and ATPase7 to 40% in ATPase 9 and AK 1.1 (Fig. 5A). The effect of RNAi was also assessed by comparing larval weight between treatments. Larvae fed on non-transgenic controls had a mean weight of 3.5 mg, while those fed on leaves of the different RNAi transgenic plants had a mean weight of 1.7–2.4 mg (Fig. 5B).

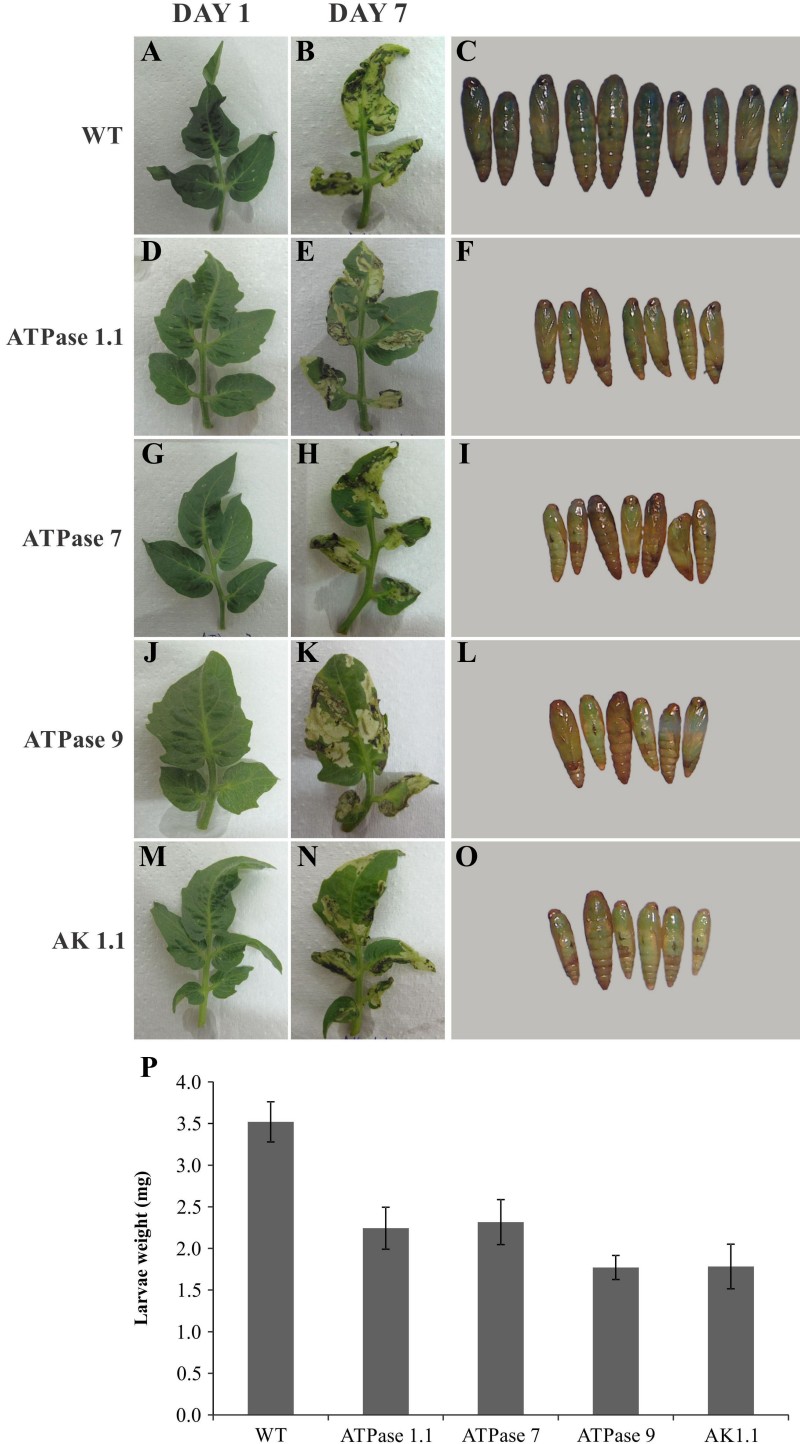

**Figure 5** **RNAi transgenic plants.** (A, B, D, E, G, H, J, K, M, N) Transgenic 'Micro-tom' tomato leaflet at one (A, D, G, J, M) and seven (B, E, H, K, N) days after *T. absoluta* larva herbivory. Pupae (C, F, I, L, O) obtained from the respective treatments. (P) Average pupae weight obtained from larvae fed on the various transgenic events, with standard error ($n = 6$–10)

Visual analysis of leaves from the different treatments revealed a clear protective effect against larval herbivory in RNAi transgenic tomato plants based on an analysis of the leaflets before (Figs. 5Aa–5Ae) and after (Figs. 5Af–5Aj) larval feeding. Whereas non-transgenic leaves had almost no undamaged leaf blade (Figs. 5Af), areas of apparently intact leaf blade were clearly seen in RNAi transgenic leaves (Figs. 5Ag–5Aj).

## DISCUSSION

The tomato leafminer *T. absoluta* can cause up to 100% damage in tomato plants in various regions and under diverse cultivation systems (*Desneux et al., 2010*; *Urbaneja et al., 2013*). Resistance to insecticides has been reported for this pest, making the development of alternative means for control even more urgent (*Urbaneja et al., 2013*; *Campos et al., 2014*). In this work, we demonstrated that RNAi for *V-ATPase* and *AK*, known RNAi target genes among pest insects, significantly reduces target gene expression, increases larval mortality and reduces leaf damage caused by larval herbivory. Both genes studied here have been previously used as RNAi target genes in studies with various agricultural pests, but this is the first report on *AK* silencing in a lepidopteran species.

In this work, we used two RNAi delivery methods to investigate gene silencing in *T. absoluta*. The 'leaf-absorbed dsRNA' delivery is based on the leaflet uptake of dsRNA transcribed *in vitro*, whereas the PITIGS is based on larval feeding on tomato leaflets containing dsRNA transcribed *in planta*. The 'leaf-absorbed dsRNA' delivery avoids cloning steps into specific vectors and allows a better control of provided dsRNA amounts in the leaves, while PITGS offers a more realistic trigger of the RNAi machinery in transgenic plants, since the dsRNA hairpin molecule will be transcribed and processed by the plant cells, as expected in transgenic plants. Noteworthy, the same gene construct can subsequently be used to transform the host plant. 'Leaf-absorbed dsRNA' delivery has been previously used for RNAi-based gene silencing in the sap-sucking *B. tabaci* (*Luan et al., 2013*). PITIGS is based on a known gene silencing delivery method (PITGS; (*Panwar, McCallum & Bakkeren, 2012*), but adapted here to present a novel plant-producing RNAi response targeting insect-specific genes.

Our work addresses the need of developing novel delivery methods to confer reliable method to provide dsRNA to insects and test the effectiveness of the target gene in conferring pest resistance or improved plant viability. The optimization of RNAi conditions requires an efficient, high throughput delivery system that addresses specificity and effectiveness of the target gene, optimal dsRNA size and dose response or phenotypic effect. Traditionally, RNAi target genes are screened and evaluated by adding dsRNA to artificial diets offered to insect larvae (*Terenius et al., 2011*; *Zhang, Li & Miao, 2012*). However, artificial diets are not readily available for many of the important pest species, including *T. absoluta* (*Urbaneja et al., 2013*), hence the need for alternative delivery methods to overcome this limitation. To date, two alternative methods for oral feeding have been used to deliver dsRNA to insects: microinjection into the hemocoel and soaking (*Price & Gatehouse, 2008*; *Gu & Knipple, 2013*). However, despite being very straightforward methods, RNAi-based pest control still relies mostly on host-insect interactions such that delivery methods based
on ingestion would be more appropriate for this use of RNAi. The final delivery of RNAi molecules in crop protection requires a delivery system to provide dsRNA continuously as a diet component to be ingested by the insects, either through transgenic plants expressing hairpin versions of target genes or by spraying dsRNA, currently a more costly option (*Katoch et al., 2013*).

Both of the RNAi delivery methods used here to target *ATPase* and *AK* were successful in larvae gene silencing, increased larval mortality and protection against herbivory. These methods have an advantage over dsRNA delivery via an artificial diet since they allow the use of natural feeding material (host plant leaves) ingested during herbivory. Notably, both delivery methods resulted in enhanced RNAi effects over time, with the greatest reduction in transcript accumulation for *V-ATPase* and *AK* observed at the latest sampling time points (Figs. 2 and 3), and larval mortality also increasing gradually in time (Fig. 3). This time-dependent RNAi response may reflect the systemic dissemination of RNAi or simply the prolonged effect of gene silencing on larval survival. Interestingly, insects may lack an siRNA signal amplification mechanism, as indicated by the apparent absence of a canonical RNA-dependent RNA polymerase (RdRP) in this phylum (*Gordon & Waterhouse, 2007*; *Camargo et al., 2015*), and this could significantly affect the source-to-signal ratio of the RNAi effects in pest insects. The 'leaf-absorbed dsRNA' delivery method also allowed us to assess the dose dependence of the RNAi response in leaf protection, with greater protection being observed in leaflets that absorbed the highest dsRNA dose tested (5,000 ng) for both target genes.

The use of fluorescently labelled dsRNA enabled the observation of the dynamics of dsRNA trajectory in the plant leaves and larvae. Upon uptake, the labelled molecules filled the mid rib (Fig. 1Ab) and then moved to lateral veins (Fig. 1Ad), as expected for soluble molecules flowing in the leaf vasculature. With the continuous absorption of labelled dsRNA, the fluorescent molecules started to accumulate in the veins and leaf lamina (Fig. 1Ad). This accumulation occurred from the marginal regions towards the blade center until the entire leaf was saturated with labelled molecules (Fig. S2). This phenomenon suggested that the RNAi molecules were initially distributed throughout the entire leaf vasculature, including the end of the vasculature tracks located at the leaf margins. Upon saturation of the vascular track, the molecules started to diffuse to the leaf lamina and until they covered the entire leaf.

We used the newly described 'PITIGS' delivery method to ask whether exogenous hairpin molecules would be processed by the plant RNAi machinery or maintained as dsRNA until ingested by the larvae. Infiltration of tomato leaves with *Agrobacterium* cells carrying GFP-expressing and -silencing cassettes resulted in exogenous gene silencing *in planta*. These results indicates that exogenous GFP-dsRNA molecules transcribed *in planta* are processed by the plant RNAi machinery, strongly suggesting that the RNAi mechanism that resulted in *eGFP* silencing in plant leaves would similarly process the insect-specific dsRNA into siRNA that would later be ingested by the larvae during feeding.

The effects of RNAi on gene silencing, larval mortality and leaf damage using both RNAi delivery methods led us to establish transgenic tomatoes plants carrying the same expression construct used in the PITIGS method. *Tuta absoluta* larvae fed on $T_0$ transgenic

'Micro-Tom' leaves weighted less, showed increased mortality and caused less damage to the tomato leaves. *ATPase* and *AK* are both required genes for metabolism and homeostasis in lepidopterans, affecting development, moulting and survival (*Qi et al., 2015*; *Li & Xia, 2012*; *Forgac, 2007*; *Kola et al., 2015*). We also observed defects on development, moulting and reduced larval weight and survival after feeding on *ATPase* and *AK* dsRNA-containing tomato leaves, suggesting that transgenesis could be a useful final delivery method for RNAi-mediated control of *T. absoluta*.

The results of this study suggest that sufficient dsRNA accumulation is required to produce gene silencing in *T. absoluta* and that a minimum dose is required before triggering RNAi (*Yu et al., 2012*). The core machinery for dsRNA-mediated gene silencing in insects requires transmembrane proteins, such as SID1, involved in dsRNA uptake and systemic spreading. Our results have been corroborated by the identification of putative orthologous *Sid-1* genes in the *T. absoluta* transcriptome analyzed by RNA-seq (*Camargo et al., 2015*), together with other core genes, such as *Dicer-like* and *Argonaute*; however, no canonical RdRP was found, an occurrence widely described for insects (*Terenius et al., 2011*; *Gu & Knipple, 2013*; *Katoch et al., 2013*; *Zhang et al., 2013*; *Scott et al., 2013*).

With the increase in *T. absoluta* resistance to currently used insecticides (*Urbaneja et al., 2013*; *Campos et al., 2014*), the application of RNAi could be a promising approach for controlling this insect. The major application of our findings is to provide alternative delivery methods for RNAi-based crop-protection technologies. In the absence of an artificial diet, the two alternative delivery systems can be used to evaluate the effectiveness of potential gene targets. We also demonstrated that transgenesis is a potentially useful final delivery method for RNAi-mediated control of *T. absoluta*. This method could provide a useful starting point for the development of alternatives to conventional pesticides.

### Funding

This work was financially supported by 'Fundação de Amparo à Pesquisa do Estado de São Paulo—FAPESP' through a Regular Grant (2011/12869-6), and a fellowship to Joni Esrom Lima (2010/11313-1). Roberto Camargo was a recipient of a CAPES fellowship, and Guilherme Oliveira Barbosa, Isabella Possignolo, Lazaro Peres, and Antonio Figueira received financial support from CNPq (Brazilian National Research Council). The funders had no role in study design, data collection and analysis, decision to publish, or preparation of the manuscript.

### Grant Disclosures

The following grant information was disclosed by the authors:
Fundação de Amparo à Pesquisa do Estado de São Paulo—FAPESP: 2011/12869-6.
fellowship: 2010/11313-1.
CAPES fellowship.
CNPq (Brazilian National Research Council).

## Competing Interests

The authors declare there are no competing interests.

## Author Contributions

- Roberto A. Camargo performed the experiments, analyzed the data, wrote the paper, prepared figures and/or tables, reviewed drafts of the paper.
- Guilherme O. Barbosa analyzed the data, prepared figures and/or tables, reviewed drafts of the paper.
- Isabella Presotto Possignolo performed the experiments, reviewed drafts of the paper.
- Lazaro E. P. Peres analyzed the data, contributed reagents/materials/analysis tools, reviewed drafts of the paper.
- Eric Lam contributed reagents/materials/analysis tools, reviewed drafts of the paper.
- Joni E. Lima conceived and designed the experiments, analyzed the data, wrote the paper, reviewed drafts of the paper.
- Antonio Figueira and Henrique Marques-Souza conceived and designed the experiments, analyzed the data, contributed reagents/materials/analysis tools, wrote the paper, prepared figures and/or tables, reviewed drafts of the paper.

## Data Availability

Raw data of the qPCR analyses has been uploaded as Supplemental Information.

## Supplemental Information

Supplemental information for this article can be found online at http://dx.doi.org/10.7717/peerj.2673#supplemental-information.

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
