# Peer review of "RNA interference as a gene silencing tool to control Tuta absoluta in tomato (Solanum lycopersicum)"

_PeerJ, doi:10.7717/peerj.2673_

## Round 0.1 · original submission · Minor Revisions

The reviewers have provided detailed points which will need to be addressed for this manuscript to be considered.

Reviewer 1 ·

Basic reporting

The authors have utilized RNA interference as a gene silencing tool to suppress the expression of V-ATPase A and Arigine kinase genes for the control Tuta absoluta in tomato. They have recorded 30-40% inhibition of target genes transcripts in the insect larvae, which in turn caused larval mortality, decrease in larval weight and deformities in larvae development. In general, the manuscript is written well and the results were presented with clarity. The introduction and background was given in a systematic manner with appropriate literature. The manuscript structure confirms to PeerJ standard. The figures are relevant, which are of high quality and well labelled and described. The authors have also supplied raw data.

Experimental design

The paper clearly define the research question, which is relevant and meaningful. The appropriate methods have been adopted to address the research problem, and such methods are described well and reproducible.

Validity of the findings

The presented data is robust and statistically sound. The data presented in a nice manner with sound conclusions, and there were interpreted meaningfully with supporting literature.

Additional comments

My general comments are:
For abbreviations, the authors should provide the elaborative form first time with its abbreviation and subsequently they should use only abbreviation consistently in the rest of the manuscript. Did authors check the selected sequences for off-target effects using in silico analysis?
Line 98 - The Panwar ...reference with multiple authors should be cited as (Panwar et al., 2012)
Line 100 & 101 - Use abbreviations.... V-ATPase & AK as the elaborative forms were already given in the beginning of the Ms.
Line 107 & 108 - Use abbreviations.... V-ATPase & AK; Avoid 'first report' as there is one report on V-ATPase in lepidopteran insect pest (Pink worm).
Line 194 - Change 'Double-stranded RNA' to 'dsRNA'
Line 207 - Mention the stage of larvae
Line 287 - Total DNA and RNA....Pl. mentioned the method that was used for DNA isolation and cite the reference for it.
Line 342- 353 -The section 'dsRNA delivery method for T. absoluta in tomato' is a repetition with Methods and it should be deleted as it doesn't contain any results and merely the description of the method again.
Line 346 - Use the abbreviation PITGS
Line 441 - References should not be cited in 'Results' as the discussion section is given separately. Delete the cited reference
Line 474 - Avoid 'first report'.
Line 485 - Change the ref. to Zhang et al., 2012
Line 500 - Delete 'first time' as VIGS approach has been used for Manduca insect
Line 654 - Terenius.....ref. No need to give the names of all the authors. Use et al. after citing first four authors.
Fig. 3 - Use % in bracket after Mortality in the Y axis, and delete % given for all the numbers
In Fig. 5B - Apply 't' test
The Discussion is more of descriptive and there is a repetition of results in detail at several places, which should be avoided. The interpretation of results in discussion should be supported with appropriate literature. There are very few references cited in the discussion. Authors should update the literature in the Introduction and Discussion sections.

Reviewer 2 ·

Basic reporting

No comments

Experimental design

1. The authors show amino acid sequence homology (Figure S1) for the two insect genes from other insect species. Do these two insect genes have any nucleotide sequence homology to the tomato genome? How did the authors minimize the possibilities of having off-target effects?

2. The authors generated T0 stable transgenic tomato lines and tested for silencing. Were T2 and T3 transgenics tested for silencing? Were there any homozygous lines for the two insect genes if tested?

3. The authors used GFP as controls in their experiments. Have the authors used empty vector controls and WT controls for the two methods described?

4. Did the authors weigh and measure the size of larvae before and after feeding at various time points in both the dsRNA delivery methods?

5. In figure S5A, the bands are not very clear. The bands in AK1.1, AK2.2, and AK2.3 look like a smear. Lanes AK5 and AK6 has more than one band. The authors should address this.

6. The authors should indicate fragment sizes in both figures S5A and S5B.

Validity of the findings

1. Does using a high concentration of dsRNA have any pleiotropic effects? This should be stated clearly.

2. The authors mention the number of larvae used for each of the dsRNA uptake delivery method. The authors should state clearly how many larvae used in their study survived up until what stage of their life cycle. How much time longer did it take for larvae survival after dsRNA feeding?

3. Was RT-qPCR conducted on the surviving larvae? If so, this should be stated clearly.

4. Have the authors infiltrated leaves (not detached leaflets) still attached to the tomato plant and used those plants to feed the larvae with dsRNA genes?

5. The authors show silencing both V-ATPase and AK insect genes have an impact on larval mortality, therefore could feeding larvae with a combination of both dsRNA genes completely destroy all the larvae?

6. Why is there no consistency with time points for any of the delivery methods used? On what basis were those time points chosen? The authors should address this.

7. The authors show confocal images of larva fed on Cy3-labeled dsRNA V-ATPase gene in figure 1B. It would be nice to also see confocal images of Cy3-labeled dsRNA AK gene in the larvae fed on tomato leaflets.

---

## Round 0.2 · accepted · Accept

Thank you for taking time to carefully address the reviewers comments.